# Towards a Novel Digital Twin Framework Proposal Within the Engineering Design Process for Future Engineers: An IoT Smart Building Use Case

**DOI:** 10.3390/s25113504

**Published:** 2025-06-01

**Authors:** Angeliki Boltsi, Dimitrios Kosmanos, Apostolos Xenakis, Periklis Chatzimisios, Costas Chaikalis

**Affiliations:** 1Department of Digital Systems, University of Thessaly, Geopolis Campus, 41500 Larissa, Greece; axenakis@uth.gr (A.X.); kchaikalis@uth.gr (C.C.); 2Department of Information and Electronic Engineering, International Hellenic University, 57400 Thessaloniki, Greece; pchatzimisios@ihu.gr; 3Department of Electrical and Computer Engineering, University of New Mexico, Albuquerque, NM 87131-0001, USA

**Keywords:** Digital Twin (DT), Engineering Design Process (EDP), engineering education, IoT, sensors and actuators, smart systems

## Abstract

The continuous evolution of Internet of Things (IoT) technologies presents significant opportunities and challenges within the domain of engineering education. This paper introduces a novel and comprehensive framework that extends the established Engineering Design Process (EDP) by incorporating a modular Digital Twin (DT) structure specifically tailored to smart building IoT applications in education. Unlike previous approaches, our framework enables real-time system feedback, simulation-based design iteration, and hands-on experimentation—all integrated within a pedagogical flow aligned with engineering curricula. It comprises seven distinct phases, providing a complete methodology that guides learners from fundamental concepts to advanced applications, including data visualization, real-time simulation, and system optimization. To demonstrate the applicability of the proposed framework, we design and experiment with a practical use case related to a meteorological station and data, which incorporate IoT-enabled sensors, actuators, and microcontrollers for real-time monitoring of environmental parameters and energy consumption within a smart building campus facility. Additionally, to support EDP extension, a hybrid pedagogical approach is introduced, which combines traditional engineering hands-on education methodologies with DT activities, to further foster experimental learning, iterative system design, and complex systems thinking development. To this end, our approach aims to bridge the gap between theoretical science and engineering knowledge, along with practical application use cases, contributing to a better preparation of future engineers capable of addressing interdisciplinary challenges associated with smart systems and digital transformation within the Industry 4.0 era.

## 1. Introduction

Universities play a pivotal role in equipping young individuals with the essential skills required for the effective utilization of technology. Nevertheless, traditional educational models frequently prioritize theoretical approaches, offering limited opportunities for students to engage in experiential learning scenarios and real-world problems. The objective of this work is to bridge the gap between theoretical science and engineering knowledge, along with practical application use cases, contributing to a better preparation of future engineers capable of addressing interdisciplinary challenges. While several studies have applied Digital Twin concepts in engineering contexts—mainly for industrial applications, predictive maintenance, or control system development—their integration into formal educational methodologies such as the EDP remains limited. Previous works tend to treat DT and EDP separately, lacking a unified structure that guides learners through iterative design using real-time system feedback. Our framework builds on these foundations by explicitly embedding DT functionality within each phase of the EDP, providing learners with a continuous, feedback-driven, and hands-on design experience grounded in real-world data and system behavior.

Our proposed use case scenarios are built around IoT sensors for remote sensing and communication technologies. The framework operationalizes the integration of theory and practice by engaging students in both conceptual analysis (e.g., interpreting environmental sensor data and understanding system behavior) and applied system design (e.g., configuring IoT hardware, implementing control logic, and simulating real-time system responses in Unity). This dual approach allows learners to move seamlessly from analytical understanding to hands-on system development, fostering both cognitive and technical skillsets. For our use case, we draw data from an experimental meteorological station developed at the department of Digital Systems, University of Thessaly. By using real-time data, students can design a smart building monitoring system, which adjusts it energy consumption demand in real time. The proposed use case centers around an experimental meteorological station. This station collects real-time environmental data using IoT-based sensors, including temperature, humidity, and light intensity. These data streams are used within the Digital Twin environment to simulate and analyze the smart building’s behavior. One key application is the adjustment of Heating, Ventilation, and Air Conditioning (HVAC) settings based on dynamic inputs, such as external weather conditions. Through this real-time data exchange, the system enables energy optimization and environment-responsive control strategies—core elements of the DT-enhanced Engineering Design Process (EDP).

As shown in [1], the meteorological station monitors environmental variables such as temperature, humidity, and wind speed while optimizing responses like shading and Heating, Ventilation, and Air Conditioning (HVAC) control based on real-time data. The integration of Digital Twin (DT) technology is justified by its capacity to create a dynamic link between the physical and virtual worlds, allowing learners to visualize, analyze, and optimize real systems in a safe and controlled manner. In contrast to traditional simulation tools, DTs enable real-time monitoring and bidirectional interaction with physical hardware, providing immediate feedback for engineering decisions. This interactivity enhances system-level understanding and supports the development of problem-solving and design-thinking skills—key competencies in the era of Industry 4.0 and Industry 5.0.

Our proposed DT framework, allows learners to experience the complexity of interconnected systems and understand the interoperability of its IoT components. IoT and edge computing technologies enhance the capabilities of DT systems by enabling seamless data acquisition and low latency communication, which are critical for real-time system synchronization. The proposed DT framework integrates these technologies to facilitate iterative improvement and real-time optimization of physical systems. In addition to the framework, a hybrid pedagogical approach is introduced to enhance engineering education. Traditional methodologies often do not provide tools for real-time feedback and iterative learning.

The key contributions of this study are outlined as follows:

1. *Design of a modular DT framework:* We propose a structured, seven-phase DT framework that supports environmental monitoring, real-time simulation, control, and system interoperability. It is specifically adapted to support engineering education by facilitating continuous feedback and iteration.

2. *Integration of emerging sensing and communication technologies in educational settings:* The framework incorporates IoT-based sensors and edge/cloud computing infrastructure, providing students with hands-on exposure to technologies shaping modern engineering practices.

3. *Testing the framework under a real-world use case:* Our DT system draws data from an experimental meteorological station, which is designed and implemented as a proof-of-concept. The station collects real-time environmental data and optimizes the building’s energy demands related to HVAC and shading control settings.

4. *Proposing an Integrated Digital Twin Engineering Process (IDTEP):* We propose a methodology which enhances EDP with DT tools and techniques, encouraging experiential, interdisciplinary learning through simulation, prototyping, and system deployment.

The framework is adaptable to multiple interdisciplinary engineering sections and is designed to support flexible and remote learning environments. Additionally, it promotes computational thinking, design optimization, and interdisciplinary collaboration, aligning with the educational needs of Education 4.0 as Industry 4.0 implies. By combining technical innovation with educational reform, this research seeks to bridge the gap between theoretical knowledge and practice, equipping engineering students with the competencies required to design, develop, and optimize smart systems in an increasingly connected and data-driven world.

## 2. Related Work

### 2.1. Applications and Use Cases of Digital Twin Technologies

Digital Twin (DT) technologies have been predominantly utilized in industrial applications, focusing on predictive maintenance, process optimization, and real-time decision-making [2]. These technologies enable seamless synchronization between physical systems and their virtual counterparts, allowing for continuous monitoring, advanced analytics, and iterative improvements. While DTs have been successfully integrated into building management systems for energy monitoring and fault detection [3,4,5], their application in the Engineering Design Process (EDP) remains largely unexplored. The EDP is a structured methodology used in engineering problem-solving, consisting of stages such as problem identification, brainstorming, prototyping, testing, and iteration [6,7,8,9,10,11,12,13,14,15]. It serves as a foundational approach in engineering education and practice, guiding the development of innovative solutions across various domains.

However, traditional EDP implementations often rely on physical prototyping and empirical testing, which can be resource-intensive and time-consuming. Digital Twin technology presents an opportunity to enhance the EDP by providing a virtualized environment where engineers and students can test designs, simulate operational conditions, and optimize performance before physical implementation. Despite the potential benefits, the literature on integrating DTs into the EDP remains scarce. Existing research has explored digital simulation tools and virtual prototyping for product design and development [16], yet these implementations often lack real-time feedback and data synchronization [17]. Studies [18,19,20,21,22,23,24,25,26,27,28,29,30,31,32,33,34,35,36] in smart manufacturing and industrial systems have demonstrated the advantages of DT-driven iterative design, enabling engineers to refine system architectures based on live performance data.

Although several studies have explored the application of Digital Twins in education and industry, many lack integration with structured pedagogical frameworks such as the EDP. For instance, existing DT-based educational tools often operate in isolated modules, without aligning simulation, data collection, and physical system interaction in a cohesive learning process. Furthermore, most reviewed works do not incorporate real-time bidirectional feedback, which is critical for understanding dynamic system behavior. These limitations highlight a gap that our framework addresses by embedding DT technologies into the full cycle of the EDP, thereby enabling experiential, iterative, and data-driven learning. In educational contexts, DTs are rarely incorporated into the iterative learning process, limiting their potential in fostering design thinking, experimentation, and system optimization. Our study extends conventional EDP methodologies by integrating real-time IoT-enabled sensor data, cloud computing, and AI-driven analytics. This enables a dynamic design loop where students and engineers can iteratively refine prototypes based on real-world insights, reducing the need for costly physical iterations.

For example, in smart infrastructure design, a DT-based approach allows for the simulation of structural integrity under varying environmental conditions, optimizing materials and energy efficiency before construction begins [37]. Despite the benefits, implementing DTs in building applications still presents critical challenges regarding scalability, integration, and cost-efficiency [38]. Similarly, in robotics engineering, DTs enable real-time virtual testing of motion control algorithms, minimizing trial-and-error processes in hardware development [39]. By embedding DTs within the EDP, this study aims to bridge the gap between theoretical design principles and practical implementation. The approach aligns with the broader vision of Industry 4.0 and Industry 5.0, where digitalization, intelligent automation, and cyber–physical systems enhance engineering workflows. Future research should explore how DTs can be further optimized for educational applications by incorporating mixed reality interfaces, AI-assisted design recommendations, and cross-disciplinary collaboration tools to enhance the engineering learning experience.

Digital Twin (DT) applications have been widely adopted in industries such as manufacturing, healthcare, and smart infrastructure, primarily for predictive maintenance, process optimization, and real-time analytics [40,41]. These applications create a virtual replica of physical systems continuously synchronized through real-time data, facilitating performance analysis and decision-making [42]. While DTs are well established in industrial applications, their integration into the Engineering Design Process (EDP) remains underexplored. The EDP is a structured methodology used in engineering education and practice, encompassing problem identification, ideation, prototyping, testing, and iteration [43]. Traditionally, this process relies on physical prototyping and empirical testing, which can be resource-intensive and time-consuming [44]. However, the adoption of DTs in EDP enables engineers and students to develop virtual prototypes, simulate operational conditions, and optimize performance before physical implementation, significantly reducing costs and time [45].

Despite its potential, research on DTs in EDP is limited. Existing studies primarily focus on virtual prototyping and digital simulation tools for product design but often lack real-time synchronization and feedback mechanisms [46,47]. Recent advancements in cloud computing and AI-driven analytics have enhanced DT tools, enabling data-driven decision-making and predictive capabilities in engineering applications [48]. AI-powered DTs can predict failures, optimize system performance, and enhance iterative design through real-time adaptive simulations [49]. Moreover, edge computing plays a crucial role in real-time DT implementation by processing data closer to the source, reducing latency, and improving system responsiveness [50]. This is particularly useful in engineering education, where students can interact with both digital and physical components to understand system behaviors dynamically. The educational integration of DTs in the EDP fosters interdisciplinary learning, combining mechanical, electrical, and computational principles [51]. Additionally, DT-driven simulations can support cyber–physical systems (CPS), allowing students to engage with interactive digital environments that mimic real-world scenarios [52]. Future research should focus on expanding DT applications in education by incorporating AI-assisted design recommendations, mixed reality interfaces, and collaborative online learning platforms [53].

### 2.2. IoT and Remote Sensing for Smart Systems

IoT sensors and actuators have been extensively used in smart systems to enhance automation and decision-making processes [16]. Research such as [18] highlights the importance of IoT in environmental monitoring, while [37] investigates the role of network communication technologies in providing low-latency communication and scalability. The use of IoT protocols ensures low-latency, high-reliability communication between DT layers [54,55,56,57,58,59,60,61,62,63]. These studies underscore the fundamental role of IoT technologies in advancing the functionality and responsiveness of smart systems, particularly in areas requiring real-time processing and efficient connectivity. This study builds on these findings by integrating IoT within the Digital Twin (DT) framework, ensuring that the system responds correctly and that there is seamless integration between physical and virtual components. IoT devices are essential for continuous data collection from the physical environment, as they feed data into digital models that can be used for analysis, prediction, and optimization [64].

Meanwhile, an advanced network enhances these capabilities by providing high-speed, low-latency communication, enabling faster, more reliable interactions between the physical and digital realms. As a result, the smart system can perform sophisticated tasks with minimal delay and higher accuracy. The integration of AI-driven Digital Twin models and IoT infrastructure is increasingly recognized as key for developing sustainable smart cities and buildings [65]. A growing body of research has explored the incorporation of IoT technologies in the educational landscape, particularly in engineering education. As smart technologies increasingly shape industry practices, it is essential for educational institutions to provide students with exposure to these cutting-edge tools. This aligns with the observations in recent studies, which point out how IoT is enabling the creation of immersive learning experiences and advanced educational systems.

For instance, the implementation of IoT-enabled tools in engineering education is discussed in [6], where IoT sensors are used in laboratory settings to simulate real-world systems and monitor physical environments. These tools allow students to engage with live data streams and explore applications such as environmental monitoring, industrial control systems, and smart grid technologies. In the context of engineering programs, such technologies bridge the gap between theory and practice, offering students the chance to test their designs in realistic scenarios [66]. This hands-on approach is especially critical in fields like mechanical, civil, and electrical engineering, where students must understand how systems behave under various conditions. Similarly, the 5G network plays an increasingly important role in educational settings, as highlighted by [19]. With the promise of ultra-low latency and high data throughput, 5G is transforming how remote laboratories and virtual simulations are conducted.

For example, ref. [67] illustrates how 5G networks can enable high-quality virtual simulations in areas like autonomous vehicle design, robotics, and smart manufacturing. Students can now remotely access complex equipment or control devices in real time, even from different geographic locations. This creates new opportunities for distributed learning and collaboration, allowing students to participate in cutting-edge projects without being physically present at the site. The application of 5G networks in educational contexts is also discussed by [68], which examines the potential for 5G to enable scalable, interactive learning platforms. These platforms, supported by 5G’s high-speed capabilities, facilitate collaborative learning, where students can work together on virtual models of smart systems or engage in real-time discussions about their findings. The paper also emphasizes how 5G allows for the seamless integration of augmented reality (AR) and virtual reality (VR) technologies, further enhancing the interactivity of engineering courses [69]. By interacting with virtual versions of real-world systems, students gain a deeper understanding of complex engineering concepts [70].

Moreover, the concept of using IoT and other advanced network technologies for the development of Digital Twin-based educational tools has been explored in works like [71]. Here, the study investigates how Digital Twins, supported by IoT, can be used as powerful educational tools in engineering fields. By creating virtual replicas of real-world systems (such as factories or transportation systems), students can experiment with different control strategies and observe the outcomes in real time. This approach not only helps in mastering theoretical concepts but also provides an opportunity to apply those concepts in practical scenarios. The synergy between IoT, network infrastructures, and Digital Twin technology creates a new avenue for smart education platforms that are dynamic, scalable, and deeply connected to industry standards. As [71] suggests, using these technologies in conjunction with Digital Twins allows for the creation of flexible and interactive learning environments, which can be accessed by students globally. This opens up the possibility of remote learning experiences, especially in disciplines where physical equipment and real-world environments are difficult to access.

By integrating these technologies, educational institutions can offer a more comprehensive and forward-thinking engineering curriculum that prepares students for the increasingly connected, real-time, data-driven world of Industry 4.0. From the implementation of IoT-enabled lab environments to the use of 5G networks for advanced simulations, the potential for these technologies in engineering education is vast, offering new ways to enhance learning and practical skill development. IoT and 5G networks play a crucial role in Digital Twin ecosystems, enhancing data acquisition, real-time analysis, and system responsiveness. IoT sensors provide real-time data streams, ensuring that digital models accurately reflect their physical counterparts [72]. Meanwhile, 5G enables ultra-reliable low-latency communication (URLLC), ensuring seamless synchronization between physical and virtual systems [73].

IoT sensors embedded within manufacturing plants, healthcare devices, and smart cities enable continuous data collection of parameters such as temperature, pressure, energy consumption, and motion [74]. These data are transmitted over 5G networks, providing real-time updates to Digital Twins and facilitating instantaneous monitoring and decision-making. Unlike traditional networks, 5G enhances DT capabilities by providing high-speed, low-latency connectivity, which is critical for applications such as autonomous vehicles, industrial automation, and telemedicine [75]. Moreover, edge computing reduces bandwidth consumption by processing IoT data locally before transmitting essential insights to cloud-based DT platforms [76]. Recent studies demonstrate the impact of IoT and 5G in smart city infrastructures, where DTs are used for real-time traffic management, energy optimization, and predictive urban planning [77]. These cyber–physical environments rely on high-speed 5G connectivity to coordinate smart grids, autonomous public transport, and intelligent buildings efficiently [78].

In educational settings, IoT-integrated DTs enhance learning experiences by providing students with access to real-time remote laboratories where they can interact with complex engineering systems without the need for physical infrastructure [79]. Despite the benefits, security vulnerabilities in IoT-5G DT systems pose significant challenges. Research highlights concerns regarding data privacy, cyber threats, and unauthorized access [80]. Ensuring the integrity and security of real-time data streams is critical for DT deployment in critical infrastructures such as healthcare and defense [81]. Potential solutions include blockchain-based security frameworks, AI-driven threat detection, and end-to-end encryption techniques [82].

In summary, while previous studies have explored Digital Twins, there is a lack of a pedagogical framework like the EDP, that is, a unified model that combines both into a coherent, iterative educational tool. The proposed framework explicitly addresses this gap by incorporating real-time data exchange, simulation, and system-level feedback into the structured phases of the EDP. This contribution is innovative in its educational orientation and focus on fusion theory, practical experimentation, and system optimization in smart building scenarios.

### 2.3. Hybrid Pedagogical Approaches

Traditional engineering pedagogy has long focused on problem-solving and physical prototyping, where students primarily engage in hands-on activities to understand engineering principles. However, this approach can overlook modern tools, such as Digital Twins (DTs), that enable iterative feedback and advanced simulation, which are vital in preparing students for the evolving landscape of smart systems and industry innovations. In this context, hybrid pedagogical approaches are emerging as key to bridging the gap between theoretical knowledge and real-world applications. These approaches combine traditional teaching with cutting-edge technologies, fostering a deeper understanding through interactive simulations, real-time system engagement, and enhanced learning environments. Hybrid pedagogical approaches that integrate Digital Twins, IoT, and AI-based simulations have revolutionized engineering learning environments [83].

Recent studies, including [39,67], have highlighted the potential of simulation tools, such as Digital Twins, in enriching conceptual understanding within engineering education. Empirical findings confirm that Digital Twin learning environments enhance engagement and improve conceptual understanding in engineering education [84]. Digital Twins create virtual replicas of physical systems, allowing students to explore and experiment with real-world environments without the need for direct interaction with physical components. This iterative approach to learning gives students the opportunity to manipulate digital models, simulate different scenarios, and observe the outcomes in real-time, thus facilitating a deeper understanding of complex engineering concepts. The hybrid pedagogical approach presented in this paper integrates these advanced technologies, particularly Digital Twins, into the learning process. This integration enables students to engage with dynamic systems in real time, making the learning experience more interactive, immersive, and reflective of actual industry practices. By leveraging DTs, students can work with live data streams and simulate the behavior of physical systems, enhancing their ability to understand and apply engineering concepts. DT-based virtual labs and simulations improve learning in engineering education across multiple studies [85,86,87,88,89,90,91,92,93,94,95,96,97,98,99,100,101,102,103,104,105].

The inclusion of IoT and 5G in these pedagogical approaches further enhances the learning experience. IoT sensors, which are embedded within physical systems, provide students with continuous streams of data that feed into digital models, creating a highly responsive learning environment. For example, IoT sensors can be used in laboratory settings to monitor and control variables in real time, offering students hands-on experience with data-driven systems. The integration of 5G networks in this context allows for ultra-low-latency communication, enabling real-time interaction between students, instructors, and the systems they are studying. This facilitates not only on-site learning but also remote learning experiences, where students can participate in simulations and interact with complex systems without being physically present.

In the context of engineering education, hybrid approaches that incorporate Digital Twins, IoT, and 5G networks are becoming increasingly important. According to [6], the use of simulation tools has significantly enhanced students’ ability to grasp complex concepts by providing them with a safe, flexible environment for trial and error. Moreover, these tools allow students to engage in hands-on learning with systems that they may not otherwise have access to, such as high-cost equipment or hazardous environments. In this way, simulation tools enable students to engage with industry-relevant systems while still being able to experiment and learn from mistakes in a virtual environment. Furthermore, hybrid pedagogies that incorporate modern technologies help to address the limitations of traditional engineering education. As [67] notes, the inclusion of Digital Twins in curriculum-based projects helps bridge the gap between theoretical knowledge and its application in real-world scenarios. These virtual replicas enable students to visualize the outcomes of various design decisions and understand how their choices will impact the overall system. For instance, in mechanical engineering, students can use Digital Twins to simulate how a machine might perform under different operational conditions by adjusting parameters in real time to test hypotheses and refine their designs.

Beyond simulation, hybrid pedagogical approaches have expanded the scope of learning by enabling collaborative, remote, and distributed education. Studies, such as [68], discuss how 5G networks facilitate real-time communication and data sharing in remote classrooms, providing students with access to advanced simulations and lab experiments regardless of their physical location. The use of 5G in education opens up new opportunities for global collaboration, where students from different parts of the world can engage in joint projects and share knowledge without the barriers of geographical constraints. The potential for hybrid pedagogical approaches to reshape engineering education is vast. By integrating technologies like IoT, 5G, and Digital Twins into the curriculum, educators can provide students with a multifaceted learning experience that bridges the divide between theory and practice. These tools not only enhance student engagement but also prepare them for the future workforce, where industry demands are driven by real-time data analysis, automation, and connectivity.

Additionally, the benefits of hybrid learning extend beyond engineering education to other technical disciplines. According to [71], the use of DTs in educational contexts helps students develop critical thinking and problem-solving skills, as they can explore different scenarios and test various hypotheses in a virtual setting. This interactive approach fosters a deeper understanding of systems thinking, which is crucial in modern engineering practices, where solutions often require the integration of multiple disciplines and technologies. In conclusion, hybrid pedagogical approaches, which combine traditional engineering pedagogy with modern technologies such as IoT, 5G, and Digital Twins, are poised to transform the landscape of engineering education. These technologies offer students the ability to engage with dynamic, real-time systems that simulate real-world environments, enhancing their ability to think critically and solve complex engineering problems. As the industry continues to evolve with emerging technologies, it is essential for educational systems to adopt these tools, ensuring that the next generation of engineers is well equipped to meet the challenges of a connected, data-driven world. DTs offer a risk-free, iterative design space where students can experiment with engineering solutions, simulate real-world behaviors, and refine their designs before physical implementation [106]. This approach enhances problem-solving skills, data-driven decision-making, and interdisciplinary collaboration.

Future developments should focus on integrating AI-driven DTs to provide automated feedback on student designs, expanding virtual learning environments with cloud-based DT frameworks, and utilizing 5G-enabled IoT simulations to create real-time interactive educational platforms [107]. By leveraging DTs, IoT, and AI, universities can create next-generation smart learning environments that align with the rapid evolution of Industry 4.0 and future engineering challenges.

## 3. Methodology

### 3.1. Digital Twin Framework

Unlike traditional Digital Twin frameworks that focus primarily on industrial monitoring or simulation-based prototyping, the proposed IDTEP architecture is uniquely tailored for educational deployment. It integrates a modular learning cycle mapped directly onto the Engineering Design Process (EDP); real-time environmental interaction through low-cost sensors and actuators; student-accessible tools (Unity3D for 3D visualization and python 3.9 for microcontroller interfacing); a unified feedback loop that combines sensing, processing, and physical actuation; and, finally, a use-case-driven curriculum that fosters iterative, hands-on design.

This convergence of real-time digital twinning with educational scaffolding constitutes the core innovation of our approach. The Digital Twin framework consists of three primary layers: data acquisition, simulation and processing, and control and actuation. These layers work together in a structured, sequential flow across seven phases to capture, process, and act upon environmental and system data. Here is how each layer and phase connects:

Data Acquisition Layer: This layer is responsible for collecting real-time environmental data from various sensors. It forms the first step in the system, where data are gathered and prepared for further analysis.


**Phase 1: Environmental Sensing**


The first phase involves setting up and calibrating environmental sensors, such as temperature, humidity, light intensity, and other relevant measurements. These sensors collect data that serve as the input for the system.


**Phase 2: Data Storage and Management**


The data collected in the first phase are then stored in a time-series database. Here, the data are organized, indexed, and prepared for real-time access or future analysis.


**Phase 3: Real-Time Data Processing**


In this phase, the raw data undergo preprocessing. Techniques like filtering and normalizing are applied to ensure data quality and consistency. The data are now ready for more advanced processing in the next layer.

The data acquisition layer forms the input for the entire system. After this phase, the data flow to the simulation and processing layer for further analysis.

Simulation and Processing Layer: This layer is where the system’s behavior is simulated and optimized based on the acquired data. Here, various algorithms and models process the data and predict how the system should respond under varying conditions.


**Phase 4: Actuation and Building Response**


In this phase, data from the data acquisition layer are used in simulations to understand how the system behaves under certain conditions. The goal is to simulate responses like energy consumption or mechanical adjustments (e.g., HVAC or shading systems).


**Phase 5: Simulation and Energy Modeling**


Further simulations are carried out to model energy consumption and system performance. Predictions about energy efficiency and system behavior are made, highlighting areas that need optimization for better performance.

The simulation and processing layer processes the data and generates results that inform the next step, which is the control and actuation layer, where actions will be taken to adjust real-world systems.

Control and Actuation Layer: This layer is where the system takes the output from the simulation and processing steps and applies it to real-world devices and systems to maintain or improve performance.


**Phase 6: Visualization and Comparison**


Data and results from the simulation phase are visualized in real-time dashboards. This helps stakeholders understand the system’s status and energy performance. Energy savings, for example, are tracked and displayed graphically.


**Phase 7: Integration and Interoperability**


The final phase ensures that the system can interact with external systems and APIs, ensuring seamless integration. Communication protocols, like MQTT, are implemented to allow different components of the Digital Twin to communicate efficiently. This phase also ensures that the system is scalable and interoperable across different platforms.

The control and actuation layer takes the processed information from the previous layer and translates it into actions, ensuring that real-world systems (like HVAC, lighting, and shading) are controlled effectively and in synchronization with the virtual model.


*Overall Workflow and Data Flow:*


The workflow of the entire Digital Twin framework follows a simple sequential circuit-like structure where each phase serves as a processing stage for the data:

The data acquisition layer collects and processes raw data from environmental sensors. The simulation and processing layer analyzes data to simulate and optimize system performance. the control and actuation layer uses processed data to control real-world actuators and visualize the system performance.

In essence, this framework ensures that data are continuously acquired, processed, simulated, and acted upon with real-time feedback loops. The system dynamically adjusts to changing conditions by optimizing performance in real time and providing users with valuable insights to drive informed decisions. Each layer builds upon the previous one, creating a robust and adaptable Digital Twin system for monitoring and optimizing environments and systems. Figure 1 above presents the modular structure of the Digital Twin framework, demonstrating how each layer functions through its distinct phases to create an optimized, synchronized system for real-time monitoring, simulation, and control.

### 3.2. Weather Monitoring Use Case for DT Framework Testing

The framework is designed with flexibility in mind, allowing it to adjust to various engineering use cases, including the meteorological station. By incorporating real-time data from the station, the system can seamlessly respond to different environmental conditions, making it suitable for a wide range of applications such as energy management or HVAC optimization. This adaptability ensures that the Digital Twin framework can be leveraged across diverse engineering studies. Recent studies have also emphasized the use of Digital Twins for enhancing both system efficiency and occupant wellness in smart environments [108]. In the following section, we will explore how the framework specifically adjusts to the use case of the meteorological station, demonstrating its capabilities in a practical application. The modular Digital Twin (DT) framework consists of three levels, each designed to support specific aspects of system design, simulation, and optimization.

Data Acquisition Layer: This layer integrates IoT sensors such as the DHT22 sensor for temperature and humidity, BH1750 for light intensity, and anemometers for wind speed. Data are collected in real time and transmitted to the DT model via 5G communication. The framework employs MQTT protocols for low-latency, reliable data transmission.Simulation and Processing Layer: The DT model, developed using MATLAB and Unity3D, simulates system behavior under varying conditions. Predictive analytics algorithms are integrated to optimize responses [109], such as shading adjustments, to reduce HVAC energy consumption. This layer provides visualizations that aid in identifying inefficiencies and refining system parameters.Control and Actuation Layer: Actuation commands are generated based on information from the simulation layer. These commands control shading systems via servo motors, adjust lighting levels, and regulate HVAC systems. Feedback loops ensure that physical responses are reflected in the virtual model, maintaining synchronization between the physical and digital systems.

These layers are divided into seven phases. The framework guides learners through these seven progressive phases, combining theoretical instruction with practical implementation.

Environmental Sensing: Learners begin by developing sensors such as the DHT22 for temperature and the BH1750 for light intensity. The goal is to understand the principles of data acquisition and sensor calibration.Data Storage and Management: Learners configure a time-series database to store environmental data. This phase introduces database design principles and querying techniques, emphasizing efficient data storage methods.Real-Time Data Processing: Using Arduino, learners implement preprocessing algorithms to filter and normalize data. Techniques such as moving averages and filters are introduced to improve data quality.Control and Actuation: Learners develop scenarios for controlling actuators based on environmental conditions. For example, servo motors adjust shading in response to solar radiation, and HVAC systems adapt to temperature changes.Simulation and Energy Modeling: Learners use tools such as MATLAB or Unity3D to simulate energy consumption, comparing performance before and after implementation. This phase highlights the impact of DT-driven optimizations on energy usage.Visualization: Dashboards are developed, allowing students to visualize real-time data and actuator states.Integration and Interoperability: Students implement MQTT protocols and 5G communication to ensure seamless interaction between system components, preparing them for scalable and interconnected systems.

### 3.3. Framework Structure

The Digital Twin framework follows a structured, progressive approach, ensuring seamless integration of real-time data, simulations, and control mechanisms. Each stage builds upon the previous one, forming a comprehensive system that enhances decision-making and operational efficiency. The overall structure and flow of the framework are schematically represented in Figure 2.

The process begins with data integration and collection, where IoT sensors, including temperature, humidity, and light intensity monitors, gather real-time environmental data. This information is sourced from an online meteorological station and is transmitted via microcontrollers, such as Arduino, ensuring accuracy and reliability. The system also establishes a communication link to efficiently retrieve and process data at predefined intervals. Once collected, the data move to data storage and management, where they undergoes preprocessing to filter out anomalies and standardize formatting. A structured database is implemented, enabling efficient storage, retrieval, and historical analysis. Cloud-based solutions ensure scalability, while backup strategies safeguard data integrity.

With a structured dataset in place, the framework advances to real-time data processing and control, where algorithms analyze environmental conditions to generate actionable insights. Based on predefined rules, the system dynamically adjusts shading mechanisms, regulates HVAC operations, and optimizes energy consumption. Communication protocols, such as MQTT, ensure low-latency transmission of control commands. Following this, the actuation and building response stage enables the execution of control strategies. Actuators, including servo motors and smart lighting systems, respond to commands by adjusting physical parameters in real time. Feedback loops continuously validate system responses, ensuring synchronization between the Digital Twin model and physical infrastructure.

To assess system efficiency, the simulation and energy modeling stage evaluates energy consumption patterns and the impact of implemented optimizations. Advanced simulation tools, such as MATLAB or Unity3D, help compare real-world data with predictive models, refining decision-making processes to enhance sustainability. Visualization plays a crucial role in system monitoring, and the visualization and comparison stage enables real-time tracking through dashboards and 3D models [110]. Predictive analytics and simulation tools such as Unity and MATLAB are integral to refining DT responses [105,111,112,113,114,115,116,117,118,119,120,121,122,123,124,125]. These tools provide intuitive representations of environmental conditions, actuator states, and energy usage trends, allowing for informed decision-making and performance assessments.

Finally, integration and interoperability ensure seamless communication and scalability across system components. Secure data exchange protocols, including MQTT and HTTP, facilitate reliable interaction between sensors, actuators, and cloud-based analytics. Compatibility testing ensures system stability, allowing integration with third-party applications and external platforms. Through this structured approach, the Digital Twin framework effectively bridges the gap between virtual simulations and real-world applications, optimizing environmental monitoring and control strategies.

The arrows in this system diagram represent the flow of data and control signals between different phases, each of which plays a crucial role in the overall process. Starting with the first phase (P1), sensor data are collected from IoT devices, such as the DHT22, BH1750, and anemometers, and are transmitted to the data storage phase (P2). These raw environmental data, which include temperature, humidity, wind speed, and light intensity, are stored for further use. In the next phase (P2 to P3), the stored data are retrieved and processed in real time for tasks such as filtering, error correction, and outlier detection. These preprocessed sensor data are formatted for analysis. In phase P3, the processed data are analyzed, and control decisions are generated to trigger actuator responses, such as adjusting shading or modifying HVAC settings.

These decisions are then passed on to the control and actuation phase (P4), where specific commands are sent to actuators, based on decision rules like “Close shades if sunlight > 800 W/m^2^”. These actuation commands are also utilized in Digital Twin simulations to optimize future responses, and the actuation history along with system performance data flows to the simulation phase (P5). Here, the results of the simulation, including energy savings predictions and efficiency reports, are sent to the visualization phase (P6), allowing for real-time monitoring through dashboards. The visualized data, which consist of energy usage trends and predictive analytics, assist in system-wide optimization and integration with external cloud services in the system integration phase (P7). This phase helps monitor system performance, alerting for any issues and providing insights for optimization. Lastly, feedback is sent back to the data collection phase (P1), where system feedback allows for adjustments to sensor configurations and data collection processes. This cycle ensures continuous improvement by updating sensor configurations and calibrating the data.

In addition to the main arrows, dotted feedback loops also play an important role. For example, feedback from actuator responses (P4 to P3) helps refine data processing algorithms, ensuring that control logic adjusts based on actuator performance, such as slow responses from shading actuators. System integration feedback (P7 to P5) helps update simulation models, improving predictions for system behavior like HVAC efficiency. Moreover, simulation feedback (P5 to P4) informs better control strategies, adjusting future decisions for things like shading adjustments based on updated logic. These feedback loops enhance the system’s overall accuracy and efficiency. In conclusion, the arrows and feedback loops define the relationships between different Digital Twin phases, illustrating how data, decisions, and control signals move through the system to optimize performance and adapt to changing conditions.

### 3.4. Weather Monitoring Use Case

The meteorological station serves as a practical application of the DT framework. Its design integrates hardware and software components to demonstrate system optimization in a real-world context. The meteorological station serves as a practical, open-ended interdisciplinary problem that directly connects to the proposed three levels of modular DT design—data acquisition, simulation and processing, and control and actuation—while also demonstrating the system’s optimization in a real-world context. This use case was selected because it effectively integrates both hardware and software components, offering students and practitioners a hands-on opportunity to address complex, real-world environmental challenges. Also, it provides a concrete example of how real-time data collection, IoT-based infrastructure, and simulation capabilities can be effectively integrated to optimize environmental monitoring and building control systems.

The technical architecture of the meteorological station, which integrates both hardware and software components, is illustrated in Figure 3. The data collection process begins with environmental sensors (DHT22 for temperature and humidity, BH1750 for light intensity, and wind speed sensors), which continuously monitor real-time weather conditions. These sensors are connected to Arduino ESP32 microcontrollers, which serve as the central processing unit for acquiring and transmitting the data. The ESP32 boards communicate via Wi-Fi, sending collected data to a cloud-based server or local database for further processing [126].

The connections in the Integrated Digital Twin Engineering Process (IDTEP) diagram are carefully designed to represent specific types of relationships, interdependencies, dependencies, and flows between the phases of the framework. Each connection plays a distinct role, working together to create a coherent and functional system. The connections with dashed lines represent the input/output data flow, where the output from one phase serves as the input for another. These connections are necessary to transmit raw data, processed information, or commands between components in a sequential manner, such as sensor data collected in the data acquisition phase being sent to the data processing phase for analysis. The triangle-shaped connections indicate feedback loops, which enable iterative improvement and synchronization between physical and virtual systems. Feedback is critical for maintaining alignment between the Digital Twin model and the real environment. For example, actuators in the building system phase send feedback to the control logic to improve predictions and ensure optimal performance. Dotted connections represent control logic or dependencies. These connections incorporate decision-making processes and control signal flow, specifying how various system components should behave under specific conditions. For example, control logic determines the behavior of actuators based on data from simulation results or environmental conditions. Heavily highlighted connections are used to indicate the data flow for analysis, that is, in the transfer of information used for assessments, such as energy consumption trends or retrieving historical data for simulations. They provide the basis for insights and comparisons that lead to optimization and decision-making. Finally, the plain black connections represent neutral or infrastructure links at the system level. These are used for fundamental communication between phases or to establish interoperability between components. For example, the connection between the Arduino and API development phases ensures that raw data collected by sensors are seamlessly integrated into higher-level protocols such as MQTT or HTTP. This type of connection does not directly affect control, feedback, or synchronization but provides the backbone for communication within the framework.

The Digital Twin is built by integrating the real-world sensor data into a simulation model that dynamically represents the physical system. The meteorological station was chosen because it allows for the demonstration of how the Digital Twin framework can be applied to dynamic, real-world systems that involve continuous data collection, simulation, and actuation. The use case satisfies all three levels of the DT design as follows:

Data Acquisition Level: The station collects real-time data from environmental sensors, such as temperature (DHT22), humidity (DHT22), light intensity (BH1750), and wind speed sensors, using Arduino ESP32 microcontrollers. These data serve as the input for further simulation and optimization. Simulation and Processing Level: The collected data are then used in Unity to simulate dynamic weather conditions. The weather simulation incorporates environmental factors such as wind, temperature, humidity, and precipitation, which respond to real-time data changes. This layer of simulation mirrors the real-world impact of environmental changes, such as the intensification of rainfall or the alteration of wind patterns. Control and Actuation Level: The system integrates actuators such as servo motors and relays for shading and HVAC control. These actuators adjust based on the simulated environmental conditions to optimize energy use and maintain comfort levels. The feedback loops from the simulation and processing layer guide these adjustments, creating a dynamic interaction between the physical and virtual systems.

The expected learning objectives for the meteorological station use case focus on providing learners with a well-rounded understanding of how to apply the Digital Twin framework to real-world engineering challenges. Learners will gain hands-on experience in integrating sensors and acquiring real-time environmental data, thereby developing a deeper understanding of how to capture and process physical parameters such as temperature, humidity, and wind speed. Through simulations and modeling in tools like Unity, learners will explore how to predict and visualize dynamic environmental conditions, allowing them to understand the impact of variables like wind and precipitation on system performance. Furthermore, students will learn to apply real-time data to control actuators such as shading systems and HVAC units, thus optimizing energy use and enhancing environmental comfort. The use case provides an opportunity to develop practical skills in system integration and data processing and in applying the Digital Twin technology across diverse engineering domains, such as environmental monitoring, energy management, and smart building design.

The system utilizes Arduino ESP32 microcontrollers to manage sensor data acquisition and actuator control [127]. Sensors such as the DHT22 sensor for temperature and humidity, BH1750 for light intensity, and wind speed sensors are employed. Actuators, including servo motors and relays, are used for shading systems and HVAC control. Figure 4, shows an example connection modeled in Tinkercad.

In this project, a dynamic weather simulation is created in Unity, where real-world weather patterns, such as temperature, humidity, and wind speed, are mirrored through scripting. The term “dynamic” refers to the system’s ability to change and respond to different environmental factors in real time. Visual effects such as clouds, rain, snow, and fog are simulated using particle systems and shaders, and these effects adjust automatically based on the weather conditions. The simulation includes smooth transitions between various weather states (e.g., clear skies, thunderstorms, or fog), ensuring a realistic representation of how the weather evolves. This dynamic interaction allows for greater immersion and can be applied to advanced environmental modeling and Digital Twin applications. Figure 5 shows the step-by-step development of a dynamic weather simulation in Unity. The tools selected for this framework were chosen with both technical capability and educational accessibility in mind. Unity3D is a game engine platform that supports 3D simulation and interactive visual feedback. Its scripting flexibility (via C#) enables real-time synchronization between virtual and physical environments, helping students visualize sensor–actuator interactions. MATLAB is used primarily for signal processing and simulation. Its control system toolbox is well suited for teaching transfer functions, modeling, and simple control logic—topics commonly found in engineering curricula. Python 3.9 is chosen for its low entry barrier and strong hardware interface libraries. Students use it to read sensor values, perform local processing, and publish data via MQTT protocols to the DT system. This toolchain ensures a balance between engineering rigor and learner usability.

The expected learning objectives for the weather monitoring use case focus on providing learners with a well-rounded understanding of how to apply the Digital Twin framework to real-world engineering challenges. By implementing a Digital Twin, we gain the ability to forecast environmental conditions, optimize control mechanisms, and improve energy efficiency in smart building systems. Learners will gain hands-on experience in integrating IoT sensors, acquiring real-time environmental data, processing sensor data through microcontrollers, and transmitting it over wireless networks. They will also develop dynamic simulations that visualize environmental changes and optimize real-world actuation systems using Digital Twin feedback loops.

Ultimately, the Digital Twin framework enables real-time monitoring, predictive analytics, and intelligent automation, demonstrating its value for environmental monitoring, energy management, and smart infrastructure optimization. Figure 6 illustrates the high-level architecture of the implemented Digital Twin framework, showcasing the interplay between the physical system, comprising Arduino ESP32 microcontrollers and environmental sensors, and the virtual component responsible for simulation, control, and data analysis. This multi-tiered approach demonstrates how real-time sensor data flow through the communication infrastructure into the simulation environment and, in turn, how optimized commands are transmitted back to actuators. The structure follows a modular, scalable design that aligns with existing IoT-DT integration strategies in smart environments.

It is worth noting that while the framework supports real-time decision-making and control (e.g., HVAC and shading systems), this work does not aim to contribute novel developments in control theory or predictive algorithms. The control logic is implemented based on rule-based or illustrative examples, serving primarily as a pedagogical use case within the broader Digital Twin framework. Figure 7 below shows the system architecture of the proposed Digital Twin use case. The diagram illustrates the flow of data from IoT sensors (DHT22 and BH1750) to the ESP32 microcontroller, which transmits the information to the cloud-based system for processing and simulation. Based on this, decisions are made and relayed back to actuators such as HVAC control or shading motors.

#### 3.4.1. The Need for a Digital Twin

The Digital Twin (DT) model does not replace the physical model but instead complements and collaborates with it. In our use case, the physical meteorological station collects real-time weather data through sensors (e.g., temperature, humidity, and wind speed), and these data are then passed to the DT model, which replicates the station’s environment in a virtual space. This integration enables the continuous monitoring, analysis, and control of the system, providing valuable insights that support decision-making. Simulation scenarios within the DT model consider various weather conditions, allowing for optimizations in shading and energy consumption. The use case evaluates key system performance metrics, such as energy savings, response times, and scalability, under different environmental conditions. By working in tandem with the physical station, the DT framework enhances system efficiency without replacing existing hardware. The available datasets, including temperature, humidity, light intensity, and wind speed, are transmitted from the physical sensors to the DT model, ensuring a seamless flow of real-time data that inform the virtual simulations and control actions.

Environmental monitoring presents significant challenges, including ensuring data accuracy, enabling real-time responsiveness, and analyzing long-term trends. Energy-aware building control via DTs is studied in [128,129,130,131,132,133,134,135,136,137,138,139,140,141,142]. A Digital Twin framework addresses these challenges by creating a virtual counterpart of the weather station, allowing for continuous monitoring, predictive analytics, and optimized resource management. Through real-time data processing and predictive modeling, the system enhances decision-making by enabling accurate forecasting and proactive response mechanisms. Resource optimization is achieved through edge processing, which reduces bandwidth consumption, while cloud scalability supports efficient long-term data storage. Real-time monitoring capabilities ensure that critical environmental parameters, such as temperature, humidity, wind speed, and precipitation levels, are continuously tracked, providing instant insights into changing conditions. Additionally, the system offers advanced simulation capabilities, dynamically modeling environmental changes to support scenario-based analysis and decision-making. By integrating intelligent automation, IoT, and digital modeling, this framework aligns with the paradigms of Industry 4.0 and Industry 5.0, improving operational efficiency and contributing to sustainable environmental management.

#### 3.4.2. Learning Objectives Through the Weather Monitoring Use Case

This framework is specifically crafted to equip future engineers with the essential skills required to excel in the evolving landscape of Industry 4.0/5.0. Through engagement with this use case, learners will gain expertise in several key areas: the integration of IoT sensors and their deployment for real-time data collection; edge and cloud computing, enabling efficient data processing and transmission; data analytics and simulation, which will allow learners to develop predictive models based on real-time data; network communication, focusing on the use of IoT protocols for seamless, low-latency data exchange; system integration, where they will learn to design and optimize comprehensive systems incorporating diverse technologies; and user-centric design principles to create intuitive dashboards for effective decision-making. This use case offers a comprehensive platform that bridges the gap between theoretical knowledge and its practical application, thus preparing future engineers to tackle the challenges and seize the opportunities presented by the ongoing digital transformation of industries.

Although this work does not focus on control optimization or algorithmic design, we include representative metrics to illustrate the system’s capabilities. These metrics include a sensor sampling rate of 1 Hz (sufficient for environmental monitoring in classroom conditions), a control loop time of ~150–200 ms from sensing to actuation, and an energy efficiency estimate, which simulated a reduction in the HVAC load by approximately 12% through real-time light/shading adjustments. These values provide students with a realistic perspective on timing and responsiveness in cyber–physical systems.

While a formal evaluation through structured testing and student feedback is part of our future work, the IDTEP framework was carefully designed to support a series of educational outcomes relevant to engineering learning. Through participation in the weather monitoring use case, students are expected to develop an understanding of how environmental sensing systems operate, how data are acquired and transmitted through IoT protocols, and how simulation tools (such as Unity3D and MATLAB) can be used to model and optimize system behavior. Additionally, learners are exposed to real-time system integration, debugging asynchronous feedback loops, and interpreting system-level behavior within a Digital Twin environment. The structured flow across the seven phases of IDTEP fosters iterative thinking, promotes awareness of trade-offs between physical and digital performance, and strengthens students’ ability to translate theoretical knowledge into practical implementation. These intended learning outcomes form the basis for planned empirical evaluations in future stages of the project.

#### 3.4.3. Framework Components

The framework consists of several key components, each playing a vital role in its operation. First, the IoT sensors at the weather station serve as the foundational element, collecting essential environmental data such as temperature, humidity, wind speed and direction, and precipitation levels. These data are critical for monitoring weather patterns, predicting precipitation, supporting renewable energy applications like wind turbines, and managing hydrological models and flood systems. These sensor inputs are then used by the Digital Twin to accurately simulate environmental conditions.

Next, the edge computing edge computing node, or the edge gateway, processes the collected data locally before transmitting it to the cloud. It helps filter and aggregate data to reduce bandwidth usage, and it supports real-time processing for time-sensitive tasks, such as triggering alerts for extreme weather conditions. This ensures system efficiency by providing low-latency responses and enabling localized decision-making. A robust communication network connects the weather station to the Digital Twin platform. By utilizing a 5G-enabled network or an IoT-based communication protocol, this network is responsible for transmitting processed or raw sensor data from the edge node to the cloud while ensuring low-latency updates of the Digital Twin. The high-speed connectivity of 5G supports seamless, continuous data flow.

The cloud-based Digital Twin platform processes and analyzes the incoming data. It stores both historical and real-time environmental data for further analysis, simulates real-world conditions by creating a dynamic digital representation, and offers predictive analytics, such as forecasting severe weather events or identifying trends. The computational power and scalability of the cloud platform make it the ideal environment for hosting the Digital Twin. Finally, the user dashboard or interface provides the point of interaction between the Digital Twin and its users. It offers real-time data visualization to display weather conditions and system states, historical data insights for analysis and trends, and decision support to assist users in making informed decisions. The dashboard is designed for ease of use, making the Digital Twin’s outputs accessible to a broad range of users, including meteorologists, researchers, and energy managers.

#### 3.4.4. System Communication and Integration Flow: From IoT Sensors to Digital Twins

The workflow begins with the IoT sensors collecting environmental data, which are processed at the edge computing node to reduce bandwidth usage and latency. The processed data are then transmitted via the communication network to the cloud-based Digital Twin platform. Here, the data are analyzed and used to simulate real-world conditions. Finally, the processed information is displayed on the user interface, providing actionable insights for decision-making. This framework demonstrates how a Digital Twin can effectively integrate IoT, edge computing, and cloud technologies to monitor and simulate weather conditions in real time. By leveraging high-speed networks like 5G, the system ensures efficient data flow and supports critical applications requiring real-time decision-making.

## 4. Integrated Digital Twin Engineering Process

Traditional Engineering Design pedagogy often follows a linear, stage-based process that emphasizes problem-solving, prototyping, and validation. However, emerging technologies such as IoT and Digital Twins (DTs) necessitate iterative testing and an interdisciplinary approach to teaching. This chapter introduces the Integrated Digital Twin Engineering Process (IDTEP), which merges traditional Engineering Design principles with DT capabilities at critical stages to enhance analysis, visualization, and real-time validation. The IDTEP constitutes a description of the data flow analyzed in the previous chapter.

As shown in recent studies, the combination of generative AI and DTs may further transform engineering pedagogy by enabling adaptive, data-driven learning systems [143]. Building on our previous work [42], this pedagogy is designed to immerse students in the development of both physical and virtual systems. By integrating DT technology into specific phases, students gain experience in modeling, simulation, and system integration, while addressing real-world constraints. The meteorological station, with its IoT and Arduino-based components, serves as a case study for this pedagogy. The hybrid pedagogical model incorporates DT tools into traditional engineering education [144]. It consists of six iterative modules:Insight Module: We define system goals and constraints using DT simulations for initial visualization. The objective is to define the engineering problem, identify constraints, and establish functional requirements. The role of the Digital Twin is to visualize environmental variables and the initial behavior of the system. In this phase, students analyze the purpose of the meteorological station, which involves collecting real-time weather data, processing them through the Arduino, and optimizing energy consumption. Tools like Unity3D or MATLAB can simulate basic environmental scenarios, allowing students to predict how temperature, humidity, and wind speed impact the system’s requirements. This early visualization helps students define performance metrics such as data accuracy, latency, and energy savings.Vision Module: We develop conceptual designs and evaluate feasibility through virtual prototypes. The objective is to create potential solutions and assess their feasibility. The role of the Digital Twin is to develop a virtual prototype of the system for initial testing. Students create multiple design ideas for integrating sensors, Arduino boards, and actuators into the meteorological station. A virtual DT model helps them evaluate the placement of motors/sensors, network topology, and actuator configurations. For example, the DT can simulate different sensor/motor placements and assess their effectiveness in obtaining accurate environmental data. In application, by using DT simulations, students can predict how the range of sensors/motors and shading logic affect the reliability of the data.Virtual Module: We model and simulate system behavior using real-time data inputs to refine designs. The goal is to create a detailed system model and simulate its behavior under various scenarios. The role of the Digital Twin is to conduct real-time simulations to test design assumptions. In this phase, students use Python 3.9 to simulate sensor data collection, actuator responses, and data communication through 5G networks. For the meteorological station, the simulations may include response times of shading actuators based on solar radiation and energy savings predicted by optimizing HVAC operations through shading. These simulations allow students to improve the system logic and identify potential bottlenecks, such as data delays or actuator latencies.Creation Module: We build physical prototypes and synchronize them with the DT model for validation. The aim is to construct the physical system and synchronize it with the Digital Twin. The role of the Digital Twin is to enable synchronization and real-time validation during the prototyping process. Students build the meteorological station using Arduino and actuators (e.g., servo motors for shading). In application, students test the shading controls by simulating various levels of solar radiation in the DT while observing the response of the physical system.Refinement Module: We use feedback from physical and virtual systems to optimize performance and address discrepancies. The purpose is to assess system performance and iteratively improve the components. The role of the Digital Twin is to identify and analyze the discrepancies between the physical and virtual systems. Using the DT, students compare simulated and actual results to identify inconsistencies [145,146]. For example, a simulation may predict a 3 °C drop in internal temperature due to shading adjustments, but physical tests only show a 2 °C drop. The DT helps students isolate factors such as sensor calibration errors or delayed actuator responses. Through this process, students iteratively refine the actuator logic and communication protocols to optimize system performance.Execution Module: We deploy the system in real-world scenarios and monitor its performance through the DT. The purpose is to develop the system in a real-world environment and validate its performance. The role of the Digital Twin is to monitor the deployed system for continuous optimization. In the final phase, students deploy the system on the campus, allowing for autonomous operation. The DT monitors system measurements, such as energy consumption, actuator response times, and environmental data accuracy. Students analyze trends over weeks to validate the scalability and effectiveness of the system. As part of the process, data collected during deployment may reveal that shading adjustments save 15% in HVAC energy, surpassing the predicted 12% from simulations.

As illustrated in Figure 8, a comprehensive overview of all the modules of the hybrid process is presented, highlighting the entire process.

The Integrated Digital Twin Engineering Process (IDTEP) is structured into six sequential modules, each representing a distinct phase of the Engineering Design and implementation cycle. These modules guide learners through a progressive development process, from problem identification to system deployment and validation. The Digital Twin is positioned as a central, integrative component that facilitates continuous interaction across all modules, enabling real-time synchronization between the physical and virtual systems. Each module represents a distinct phase in the engineering process, beginning with the identification of problem requirements and culminating in the deployment and validation of the final system. The process is structured to facilitate a progressive and systematic approach, guiding learners through successive stages of problem-solving and system development.

At the core of this framework lies the Digital Twin, which acts as a central integrative component, establishing continuous interaction between the physical and virtual representations of the system [147]. The Digital Twin serves as a dynamic repository and processing unit, enabling real-time data exchange, simulation, and system monitoring across all phases of the process. As illustrated in the diagram, bidirectional arrows connect each of the modules to the Digital Twin, reflecting the ongoing synchronization between virtual simulations and physical implementations. These connections enable iterative validation, allowing design decisions to be informed by both predictive models and real-world data.

The Insight Module initiates the process by defining the problem requirements, which are immediately integrated into the Digital Twin to ensure alignment with system objectives. Subsequently, the Vision Module focuses on developing initial design concepts by utilizing the Digital Twin to simulate various design alternatives and evaluate their feasibility. The Virtual Module further advances the process by simulating system behavior in a virtual environment, where the Digital Twin provides a real-time simulation platform to test and validate system responses under different scenarios.

Following virtual validation, the Creation Module supports the construction and synchronization of the physical system with its digital counterpart. This synchronization enables seamless data transfer between the two representations, ensuring that the physical system reflects the most current virtual model. The Refinement Module then facilitates rigorous testing and optimization, with the Digital Twin capturing real-time data from physical tests and informing iterative design improvements. Finally, the Execution Module encompasses the deployment and validation of the completed system. During this phase, the Digital Twin continues to monitor system performance, enabling ongoing feedback, adaptation, and future enhancements. The inclusion of continuous bidirectional communication between each module and the Digital Twin highlights the framework’s emphasis on iterative learning and adaptive engineering. By facilitating real-time synchronization and feedback, the IDTEP framework enhances the development of engineering competencies, promotes interdisciplinary collaboration, and equips future engineers with the skills required to manage complex, cyber–physical systems in the context of digital transformation.

### 4.1. The Circular Flow of the IDTEP

The framework of the Integrated Digital Twin Engineering Process (IDTEP) offers significant advantages, particularly within the context of modern engineering education. This framework utilizes the concept of digital twins, which are virtual replicas of physical systems, to enhance learning experiences. The primary benefit of the IDTEP lies in its ability to bridge the gap between theoretical concepts and real-world practical applications. The IDTEP promotes improved learning through simulation and visualization, allowing learners to interact with the digital representations of the physical systems they study. This enables them to observe the outcomes of their decisions and design changes in real time, deepening their understanding of the subject. Students can experiment with simulations of real-world engineering challenges by creating and testing various solutions that foster active learning and help them grasp complex engineering concepts through practical engagement. Compared to traditional teaching methods, which rely heavily on textbooks, lectures, and static case studies, the IDTEP allows students to visualize and interact with dynamic systems, significantly enhancing their comprehension and interest.

Furthermore, it provides real-time feedback and facilitates iterative learning by enabling students to observe the immediate results of their design decisions and refine their models and systems in successive iterations. The IDTEP naturally supports interdisciplinary collaboration by integrating various engineering disciplines, such as mechanical, electrical, and software engineering, into its framework, reflecting real-world teamwork on complex engineering projects. It also emphasizes flexibility and personalized learning by allowing students to adapt their learning paths, experiment with different designs, and explore diverse scenarios at their own pace. By encouraging holistic understanding, systems thinking, and design optimization, the IDTEP equips students with critical decision-making and problem-solving skills. Additionally, it is cost-effective and accessible, reducing the need for physical prototypes and allowing learning to take place in remote or hybrid settings. By leveraging digital twins, the IDTEP provides an innovative and effective approach to preparing students for the complexities of modern engineering challenges, aligning with contemporary pedagogical theories and practices.

### 4.2. Advantages of the IDTEP Framework

The Integrated Digital Twin Engineering Process (IDTEP) offers several advantages. It promotes iterative learning, allowing students to continually refine their designs using feedback from both physical prototypes and virtual simulations. The integration of Digital Twin technology enables students to monitor and analyze systems in real time, which enhances their understanding of the dynamic behavior of the system. The methodology is scalable and can be applied across various engineering fields and projects, ranging from environmental monitoring to smart infrastructure. Furthermore, by incorporating IoT and 5G technologies, students are better prepared for the modern challenges faced in engineering. The framework of the Integrated Digital Twin Engineering Process (IDTEP) offers significant advantages, particularly within the context of modern engineering education. This framework utilizes the concept of digital twins, which are virtual replicas of physical systems, to enhance learning experiences. The primary benefit of the IDTEP lies in its ability to bridge the gap between theoretical concepts and real-world practical applications. The IDTEP promotes improved learning through simulations and visualization, allowing learners to interact with the digital representations of the physical systems they study. This enables them to observe the outcomes of their decisions and design changes in real time, deepening their understanding of the subject. Students can experiment with simulations of real-world engineering challenges by creating and testing various solutions that foster active learning and help them grasp complex engineering concepts through practical engagement.

Compared to traditional teaching methods, which rely heavily on textbooks, lectures, and static case studies, the IDTEP allows students to visualize and interact with dynamic systems, as demonstrated in studies by Monostori et al. on smart manufacturing systems utilizing digital twins [71]. Furthermore, it provides real-time feedback and facilitates iterative learning by enabling students to observe the immediate results of their design decisions and refine their models and systems in successive iterations. This aligns with the constructivist learning theory by Piaget, which emphasizes iterative engagement, exploration, and feedback as essential components of learning [148].

The IDTEP naturally supports interdisciplinary collaboration by integrating various engineering disciplines, such as mechanical, electrical, and software engineering, into its framework, reflecting real-world teamwork on complex engineering projects. This is consistent with findings by Linn et al., who advocate for the integration of multiple disciplines in engineering education to better prepare students for real-world challenges, as well as project-based learning approaches that underscore the value of interdisciplinary collaboration in fostering practical skills and teamwork [149]. It also emphasizes flexibility and personalized learning by allowing students to adapt their learning paths, experiment with different designs, and explore diverse scenarios at their own pace [150].

This adaptability mirrors personalized learning frameworks that have been shown to improve outcomes, as supported by the work of Van Merriënboer and Kirschner, which highlights the benefits of giving students control over their learning process [151]. By encouraging holistic understanding, systems thinking, and design optimization, the IDTEP equips students with critical decision-making and problem-solving skills. This systemic approach aligns with the findings by Woods et al., who emphasize the importance of systems thinking in engineering education for analyzing and designing complex systems [152]. Additionally, the IDTEP is cost-effective and accessible, reducing the need for physical prototypes and allowing learning to take place in remote or hybrid settings. Its cost-efficiency is highlighted in studies by Böhme et al., who demonstrated how digital learning environments can reduce operational costs while maintaining or enhancing educational outcomes [153]. By leveraging Digital Twins, the IDTEP provides an innovative and effective approach to preparing students for the complexities of modern engineering challenges, aligning with contemporary pedagogical theories and practices.

### 4.3. Comparison of the IDTEP with Other Frameworks

The Integrated Digital Twin Engineering Process (IDTEP) represents an innovative approach to engineering education, addressing key limitations in traditional and modern pedagogical frameworks. Below is a comparative analysis highlighting its advantages. The selection of the educational models used for comparison in this study is based on the findings of a previous survey conducted by our research team [68]. This survey analyzed prevalent learning methodologies within engineering education, identifying traditional pedagogy, project-based learning (PBL), flipped classroom, and e-learning as key pillars that currently guide the instructional strategies for engineering students. These models serve as foundational reference points in shaping how engineering knowledge and skills are transmitted. The following comparative analysis highlights the advantages of the IDTEP approach over these established educational frameworks.

Traditional engineering education relies heavily on lecture-based learning, emphasizing theoretical knowledge and foundational principles. Practical exposure is provided through physical labs and experiments, with a fixed, linear learning path. However, this approach often lacks real-time interaction, leading to limited student engagement. It is also resource-intensive due to the reliance on physical prototypes and lacks interdisciplinary collaboration [154]. The IDTEP enhances learning by offering real-time feedback through simulations, fostering interdisciplinary collaboration, and reducing costs associated with physical models and prototypes. It allows students to engage with complex, multi-domain systems in a dynamic and interactive environment.

Project-based learning emphasizes teamwork and problem-solving through real-world challenges. While it promotes active learning, PBL faces limitations, including delayed feedback, resource constraints due to prototype development, and rigid timelines that hinder iterative learning [155]. By integrating Digital Twins, IDTEP enables immediate feedback, reducing the dependence on physical prototypes. It supports continuous iteration and flexible learning timelines, enhancing the depth and adaptability of student projects.

The flipped classroom approach prioritizes in-class active learning by reversing the roles of lectures and homework. While it encourages student engagement, this model often lacks hands-on experience and real-time simulation capabilities. Its success is heavily dependent on students’ engagement with pre-class materials [156]. The IDTEP complements flipped classrooms by incorporating interactive simulations and Digital Twins into activities, fostering deeper engagement and system-level thinking. This enhances the application of theoretical concepts to practical engineering challenges.

E-learning is characterized by flexibility and accessibility, offering modular online content such as videos and quizzes. Despite its advantages, it often suffers from limited interactivity, engagement challenges, and the absence of holistic system integration [157]. The IDTEP provides an immersive and interactive experience by leveraging Digital Twins for real-time feedback and system integration. It supports interdisciplinary collaboration, addressing the fragmented nature of traditional e-learning platforms.

The IDTEP offers significant advantages over traditional pedagogies by integrating real-time simulations, fostering interdisciplinary collaboration, and enabling flexible, resource-efficient learning. Its innovative use of Digital Twins positions it as a transformative framework for engineering education (see Table 1).

In summary, the IDTEP framework demonstrates significant benefits compared to traditional teaching methods, PBL, flipped classrooms, and e-learning. It excels in fostering interactivity, providing real-time feedback, promoting cost-effective solutions, and encouraging interdisciplinary collaboration. Through the integration of Digital Twin technology, the IDTEP cultivates a dynamic educational environment that supports system-oriented thinking, iterative design processes, and active engagement with real-world scenarios. This approach aligns with contemporary educational principles focused on experiential and active learning, equipping students with the skills to navigate complex engineering challenges effectively.

A number of studies in the recent literature have contributed valuable insights into the development and application of Digital Twin (DT) technologies, either in educational or industrial contexts. However, a detailed comparison reveals that while these works address selected aspects, none of them present an integrated framework that simultaneously combines both pedagogical and technical dimensions in the way our proposed system does. For example, refs. [2,5,18] focus on industrial or infrastructure-related implementations of DT. Specifically, ref. [2] applies DT in the context of predictive maintenance in Industry 4.0 environments, while [5] explores DT applications for energy control and HVAC optimization. Similarly, ref. [18] evaluates communication protocols and transmission technologies for sensor-to-twin data flow. These works offer robust IoT frameworks and real-time data exchange architectures but do not address educational objectives, pedagogical strategies, or integration into learning design. They also do not engage with modular or adaptive educational structures.

Reference [37] provides a virtual laboratory setup that allows remote interaction with physical systems through 3D visualization and internet-based control. While this study incorporates a complete DT architecture and physical–digital integration, it lacks a pedagogical design approach or connection to engineering education processes such as the EDP cycle. Similarly, ref. [45] investigates real-time synchronization and model updating in DT systems, contributing valuable insights into system continuity and architecture but without an explicit focus on learners or didactic strategies.

On the other hand, refs. [42,51,53] engage directly with educational implementations. Reference [42] proposes a methodology for capstone projects using MOOCs, emphasizing project-based learning but lacking a unified DT infrastructure. References [51,53] explore DT in multidisciplinary and mixed-reality learning environments, respectively, highlighting student engagement and cross-disciplinary potential. However, these studies typically omit structured integration of the Engineering Design Process and do not offer an adaptive framework that supports data-driven iteration and evaluations.

Reference [47] introduces AI-enhanced decision support in DT environments and includes MPC modeling elements, but its orientation is towards optimization rather than learning. Likewise, ref. [50] incorporates edge computing and ESP32 microcontrollers for localized data processing, contributing to system performance but without a focus on educational deployment or learner interaction. Finally, refs. [37,45] demonstrate physical computing and synchronization but fall short in adopting hands-on, student-centered learning approaches or scaffolding through pedagogical models.

In comparison, our proposed framework offers a unique combination of all the above dimensions. It adopts an extended Engineering Design Process specifically tailored for higher education, enriched with hands-on, project-based activities. It employs a modular and adaptive architecture that integrates real-time IoT data, open hardware (ESP32), and physical computing components while also ensuring structured DT–user interaction. Moreover, the framework is explicitly grounded in a pedagogical rationale, making it suitable for engineering education and aligned with contemporary didactic methodologies. This holistic integration across technical and educational domains is, to the best of our knowledge, not present in any single work reviewed.

The selection of related references was guided by stringent criteria to ensure thematic relevance and highlight the distinct contribution of the present work. Primarily, studies explicitly addressing Digital Twins (DTs) were chosen, particularly those employing IoT technologies, sensors, and actuators or those focusing on applications within education, building systems, environmental monitoring, or control frameworks. Conversely, references of a general or introductory nature, such as those broadly discussing Industry 4.0, were excluded, as were studies specialized in domains outside the scope of this research, for example, purely biomedical DTs. Additionally, references serving solely as background material without contributing to comparative analysis within the related work section were omitted.

This selection strategy aimed to comprehensively cover all thematic axes pertinent to the present contribution, including DTs for energy and building systems, pedagogical applications of DTs, IoT frameworks leveraging real-time data, and DT implementations incorporating artificial intelligence, which serve as a contrast to the current approach. Finally, the chosen studies were selected to clearly demonstrate how this work differentiates itself: it presents an educational framework that is neither AI-based nor industrial but is instead modular, hands-on, and tightly integrated with the Engineering Design Process (EDP). The selected references maintain sufficient similarity to establish relevance while differing enough to underscore the innovation of the proposed approach.

To further support the positioning of the proposed framework within the current landscape of Digital Twin (DT) educational research, Table 2 below provides a feature-based comparison of selected studies drawn from references [2,3,4,5,6,7,8,9,10,11,12,13,14,15,16,17,18,19,20,21,22,23,24,25,26,27,28,29,30,31,32,33,34,35,36,37,38,39,40,41,42,43,44,45,46,47,48,49,50,51,52,53,54,55,56,57,58,59,60,61,62,63,64,65,66,67,68,69,70,71,72,73,74,75,76,77,78,79,80,81,82,83,84,85,86,87,88,89,90,91,92,93,94,95,96,97,98,99,100,101,102,103,104,105,106,107]. This structured comparison focuses on the presence or absence of key attributes relevant to the implementation of DTs in pedagogical settings. The criteria include the integration of the Engineering Design Process (EDP), real-time data handling, physical–digital coupling, modularity, open hardware adoption, and user interaction. As shown, only a limited number of studies attempt to address more than a few of these dimensions simultaneously. Notably, the proposed framework uniquely satisfies all listed criteria, including hands-on learning, EDP alignment, modularity, and pedagogical intent, underscoring its value as both a practical and educational tool in engineering curricula. This multidimensional comparison highlights the novelty and completeness of our approach in combining theoretical design processes with real-time, sensor-driven, interactive learning experiences.

## 5. Discussion and Conclusions

This research lays the groundwork for a comprehensive framework that integrates IoT, 5G, and Digital Twin technologies into engineering education. The proposed framework aligns with core aspects of Industry 4.0, including the integration of cyber–physical systems, real-time sensor networks, edge computing (via microcontrollers), and digital simulation environments (Unity3D). Additionally, by enabling dynamic interaction between the physical and digital layers, the system embodies Digital Twin principles in a tangible, scalable educational form. It also reflects the values of Industry 5.0 and Education 4.0, which emphasize human–machine collaboration, personalization, and interdisciplinary problem-solving. By placing students in the control of full-cycle system design, the framework supports personalized, hands-on learning and promotes critical skills such as system thinking, data analysis, and control logic development.

Despite the successful integration of the Digital Twin framework, several challenges were observed during prototyping and pilot deployment:
Sensor accuracy: Environmental sensors (e.g., DHT22) are sensitive to placement and calibration and may show non-negligible drift under varying humidity and temperature conditions.Connectivity: Wi-Fi signal strength and router stability can affect data flow, leading to latency spikes or temporary disconnections.System scalability: Scaling the framework to large classrooms requires the careful management of IoT resources, microcontroller availability, and concurrent network bandwidth.Debugging complexity: Students often encountered difficulties in tracing delays or failures in the sensor–actuator communication chain, especially in real-time feedback loops.

These observations will inform improvements in robustness, tool documentation, and support material in future iterations.

Beyond the technical issues discussed above, the practical deployment of the IDTEP in real-world classrooms involves several additional challenges:Tool familiarity: Students and instructors may have limited experience with platforms such as Unity3D, requiring orientation sessions or tutorials.Instructor training: The effective use of DT-based curricula requires faculty development, including the ability to troubleshoot IoT-device setups and interpret simulation behavior.Hardware constraints: Classrooms with limited access to Arduino kits, sensors, or internet infrastructure may face difficulties replicating the full system.Time and curriculum fit: Integrating multi-phase DT projects into existing engineering syllabi may require adjustments to course timelines and assessment strategies.

We view these not as blockers but as practical considerations that must be addressed for broader adoption.

In the coming stages, the modular DT framework will continue to be refined for better system optimization and reliability. The scalability of the framework makes it suitable for various applications, ranging from smart buildings to renewable energy systems. Furthermore, the hybrid pedagogical model has proven to be a valuable tool for enhancing students’ problem-solving abilities and interdisciplinary skills. Future work will explore further directions to expand the framework’s impact across diverse engineering domains and improve the tools used to teach the next generation of engineers. By continuing to leverage IoT and 5G technologies, we intend to equip students with the knowledge and skills they need to tackle increasingly complex engineering problems in real-world settings.

The next phase of this research will focus on the practical implementation of the proposed Digital Twin (DT) framework, ensuring its effectiveness in real-world scenarios. A key component of this implementation will involve evaluating user acceptance and learning outcomes through structured assessments, including questionnaires and statistical analysis. The goal is to measure how effectively the framework supports learning objectives and assess the degree of adoption among students and instructors. We intend to incorporate structured statistical evaluations, including the use of RMSE (root of the mean of the square of errors), MAE (mean of absolute value of errors), and variance analysis, on environmental data, as well as educational impact assessments through methods such as *t*-tests or pre/post comparative studies. This will help validate both the performance of the system and its pedagogical value in controlled learning environments. To achieve this, a systematic evaluation methodology will be developed, incorporating pre- and post-test assessments, surveys, and qualitative feedback. Statistical methods such as *t*-tests and ANOVA will be employed to analyze improvements in students’ understanding of IoT, Digital Twin technology, and edge computing. Additionally, benchmarking techniques will be used to compare the framework’s effectiveness against other digital learning models. This process will help identify strengths and areas for improvement, ensuring that the framework remains adaptable and relevant for diverse educational applications.

Beyond evaluations, future work will explore the scalability of the framework across multiple engineering disciplines. While the current use case focuses on a meteorological station, the modular nature of the system allows for its seamless integration into courses covering automation, smart infrastructure, and real-time system monitoring. Expanding the framework into different academic contexts will provide students with a broader interdisciplinary learning experience, reinforcing the applicability of Digital Twin technology in real-world problem-solving. To further enhance the learning experience, interactive visualization tools will be developed. These tools will provide real-time insights into system behavior, allowing students to engage with live data, analyze trends, and gain hands-on experience. By incorporating intuitive dashboards and predictive analytics, learners will be able to better understand the impact of their decisions within a Digital Twin environment.

The contributions of this work lie in the development and demonstration of a modular DT framework for educational purposes, the integration of real-time environmental data, and the validation of pedagogical effectiveness via use case implementation. These aspects distinguish the present study from prior studies focused solely on technical or industrial Digital Twin applications. Future work will include not only technical expansion but also structured educational impact assessments using quantitative metrics and benchmarking against alternative learning methodologies.

Additionally, continuous student feedback and the iterative refinement of the framework will be prioritized. Surveys and structured interviews will be conducted to assess its usability, engagement, and overall effectiveness. Insights gained from this feedback will guide future refinements, ensuring that the framework evolves in response to the needs of learners and instructors. Ultimately, by focusing on implementation, statistical validation, and benchmarking, future work will position the framework as a scalable and adaptable model. Its structured evaluation will contribute to the broader understanding of how Digital Twin technology can enhance engineering education, making it a valuable reference point for future pedagogical innovations.

## Figures and Tables

**Figure 1 sensors-25-03504-f001:**
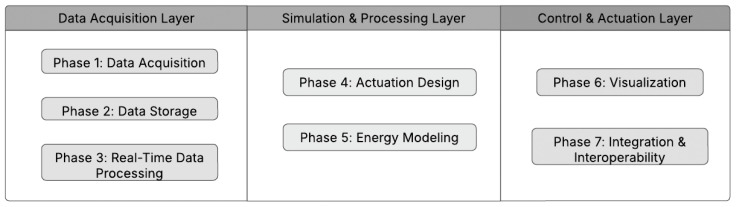
Layered structure of the proposed Digital Twin framework, highlighting the three core layers: data acquisition, simulation and processing, and control and actuation.

**Figure 2 sensors-25-03504-f002:**
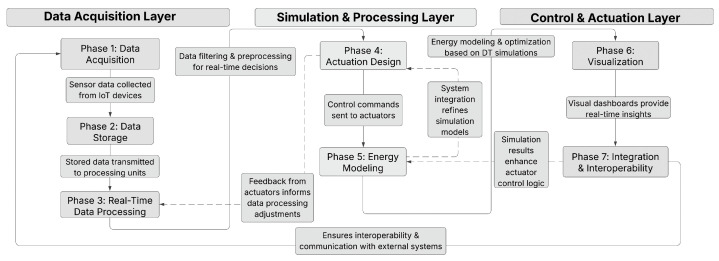
Overview of the modular Digital Twin framework, illustrating the seven phases: data integration, storage, real-time processing, actuation, simulation, visualization, and system integration. They enable the real-time monitoring, control, and optimization of building systems.

**Figure 3 sensors-25-03504-f003:**
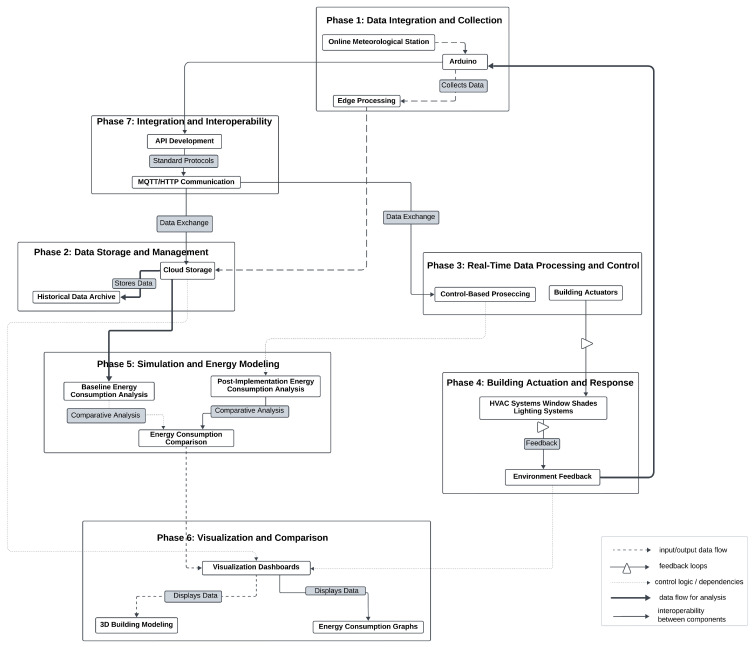
Technical architecture of the meteorological station use case: integration of IoT sensors, microcontrollers (ESP32), and communication protocols in the Digital Twin framework.

**Figure 4 sensors-25-03504-f004:**
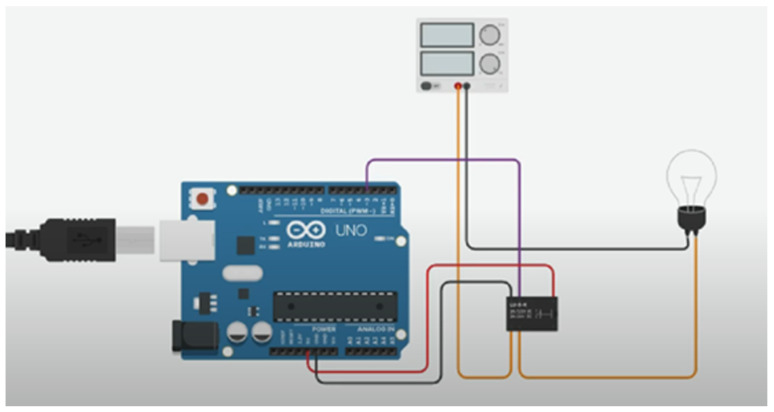
Example connection diagram of the meteorological station, illustrating the integration of sensors, actuators, and Arduino ESP32 microcontroller, as modeled in Tinkercad.

**Figure 5 sensors-25-03504-f005:**
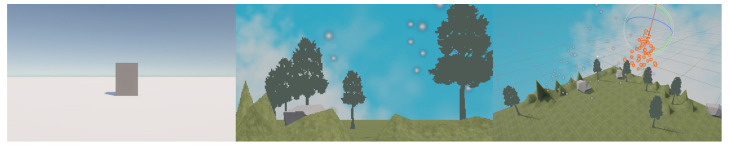
Step-by-step development of a dynamic weather simulation in Unity. The first image shows the initial lighting setup with shadows cast by the sun. The second image introduces cloud formation and light rainfall, simulating early weather changes. The third image displays more complex weather phenomena, including intensified rainfall, fog, and wind effects, highlighting the system’s ability to adjust to varying environmental conditions.

**Figure 6 sensors-25-03504-f006:**
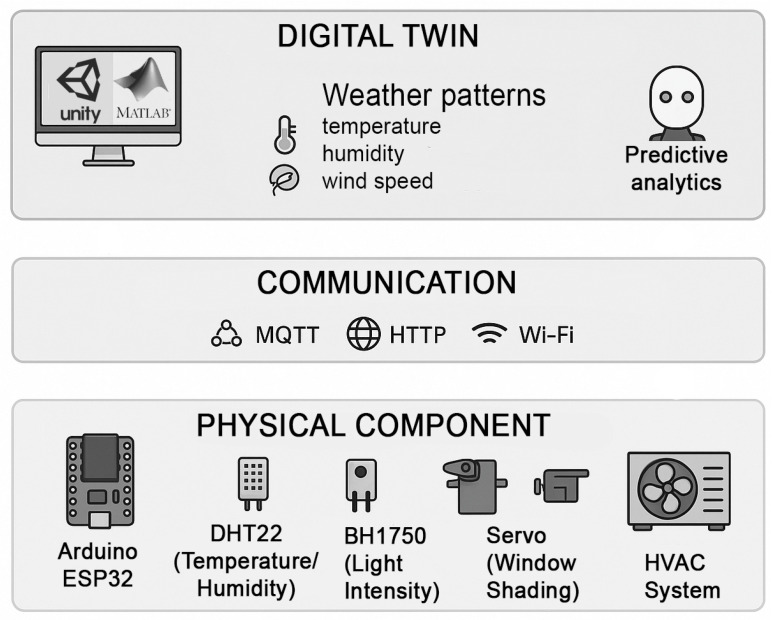
A three-part schematic representing the interaction between the physical (Arduino-based) system, the communication bridge, and the virtual Digital Twin environment. Real-time data exchange is enabled via Wi-Fi/MQTT protocols, facilitating responsive control and visualization.

**Figure 7 sensors-25-03504-f007:**
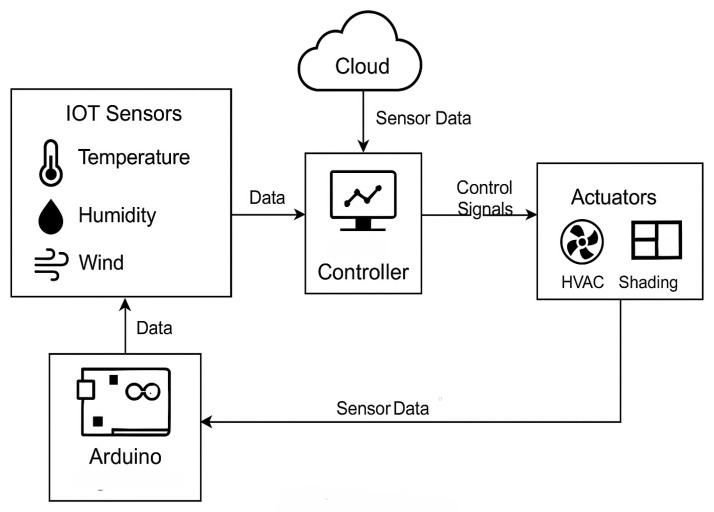
System architecture of the weather monitoring Digital Twin use case, showing the data and control flow between IoT sensors, the ESP32 edge device, cloud processing, and physical actuators.

**Figure 8 sensors-25-03504-f008:**
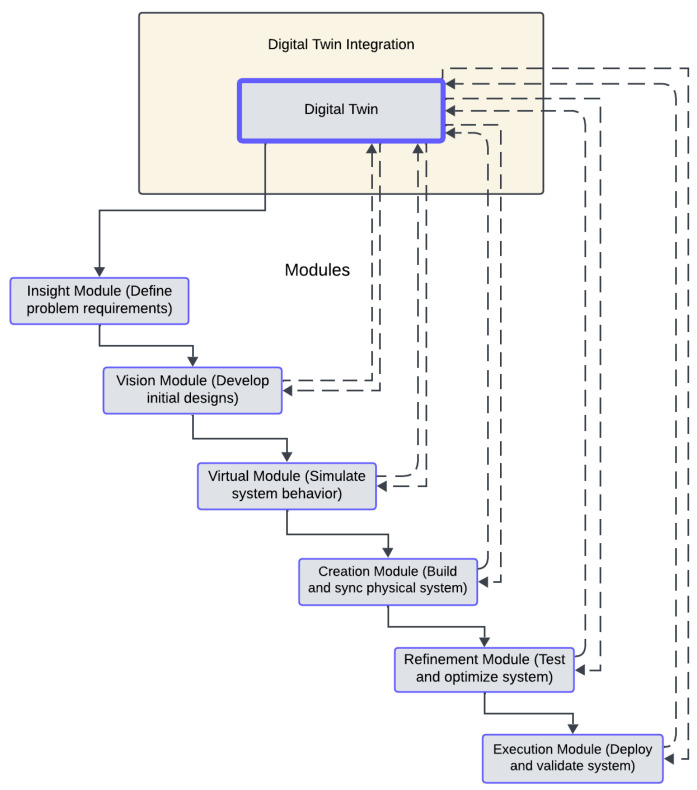
A detailed depiction of the phases involved in the hybrid process, illustrating the complete process from conceptualization to real-world deployment.

**Table 1 sensors-25-03504-t001:** The table compares various educational approaches—IDTEP, traditional pedagogy, project-based learning, flipped classroom, and e-learning—across five key aspects: real-time feedback, hands-on experience, interdisciplinary collaboration, cost, and iteration.

Aspect	IDTEP	Traditional Pedagogy	Project-Based Learning	Flipped Classroom	E-Learning
**Real-Time Feedback**	Immediate, from Digital Twins and simulations	Limited; feedback is often delayed	Delayed, based on project milestones	Minimal; feedback from class activities	Minimal; feedback from assessments
**Hands-On** **Experience**	Interactive simulations of complex systems	Physical labs and models	Hands-on projects but resource-intensive	Limited hands-on experience due to digital focus	Limited interaction with physical systems
**Interdisciplinary** **Collaboration**	High; integrates multiple engineering domains	Low; often focused on specific disciplines	Moderate; within project teams	Low; generally individual work	Low; isolated learning modules
**Cost**	Cost-effective (no need for physical prototypes)	High due to physical materials and prototypes	High due to resources for physical projects	Lower cost for resources but still limited hands-on experience	Low; lacks full engagement with complex systems
**Iteration**	High; students can iterate designs and test outcomes continuously	Limited iteration; based on fixed projects	Moderate; dependent on project timelines	Moderate; learning is focused on class interaction	Low; limited to quizzes and assignments

**Table 2 sensors-25-03504-t002:** Comparison of selected related works and the proposed framework across two thematic pillars: educational methodology and practical DT implementation.

Ref.	Education Technology and Contemporary Didactics						DT Practical Use Case		
	Educational Technology	EDP Extension Proposal	Hands-on Related Activities	Integrated DT Architecture	Use of Pedagogical Approach	Adaptive Framework	IoT Scenario	Physical Computing Tools	DT–User Interaction
[2]		✓		✓			✓		
[5]		✓		✓			✓		
[18]		✓		✓			✓		
[37]	✓		✓	✓	✓	✓	✓	✓	✓
[42]	✓	✓	✓		✓	✓			✓
[45]	✓			✓	✓		✓	✓	
[47]		✓		✓			✓		
[50]		✓		✓		✓	✓	✓	
[51]	✓	✓	✓	✓					✓
[53]	✓		✓	✓	✓				✓
Proposed approach	✓	✓	✓	✓	✓	✓	✓	✓	✓

## Data Availability

Data are contained within the article.

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
