# Peer review of "Towards a Novel Digital Twin Framework Proposal Within the Engineering Design Process for Future Engineers: An IoT Smart Building Use Case"

_sensors, 2025, doi:10.3390/s25113504_

Round 1
Reviewer 1 Report
Comments and Suggestions for Authors
- I suggest that the paper be improved in the following way: make a detailed mathematical elaboration of the MPC model and algorithm (include the state matrix, constraint conditions, discretized model form, etc.), include a system diagram, statistical robustness is lacking, improve the titles of the figures because they are short and uninformative, improve the language and expand the literature with the latest sources from reputable journals.
- As for the conclusion, it is too general, you should emphasize your contribution more clearly and suggest specific directions for further research.
- The literature used is superficial and needs to be revised and expanded with the latest sources from reputable journals.
The language is understandable, but there are minor linguistic and stylistic errors.
Reviewer 2 Report
Comments and Suggestions for Authors
- Abstract : While authors mention a “novel and comprehensive framework,” authors don’t clearly spell out how it is novel compared to extsing works.
- Introduction : Author's statement “combining theoretical learning with practical applications” are general and they need concrete elaboration.
- Introduction : Authors need to justify the use of Digital Twin (DT)
- Introduction : Some awkward or unclear phrasing like “Our proposed use case scenarios built around IoT sensors…” (missing a verb; should be “are built around”).
- Introduction : There’s no discussion of how the proposed work builds on prior DT or EDP efforts. Without that, the introduction floats in isolation.
- Introduction : Sentences like lines 41–49 pack too many ideas together (DT + meteorological station + HVAC + real-time response) without giving each piece enough clarity.
- Related works : The section mostly describes what other papers did but doesn’t critically analyze their limitations or gaps.
- Related works : While authros mention that prior work hasn’t addressed DT in the EDP much, you don’t explicitly position your study as filling this gap.
- Methodology: The description of the layers (Data Acquisition, Simulation and Processing, Control and Actuation) is repeated multiple times in slightly different wording (e.g., in sections 3.1, 3.2, and 3.2 Framework Structure), which can make the section less sharp.
- Methodology : The section present a general framework description that could apply to many Digital Twin (DT) setups. There’s little emphasis on what is new, unique, or innovative about this implementation, which is critical in a research paper.
- Methodology : For use case please Include quantitative performance metrics: For example, provide data on sensor sampling rates, network latency, control loop response time, or energy savings from optimization.
- Authors mention statements like “this framework aligns with Industry 4.0 and 5.0 paradigms” are broad and not tied to specific mechanisms or outcomes in the use case. There’s no mention of potential technical or operational limitations, such as sensor accuracy, connectivity issues, or system scalability bottlenecks. Without this, the discussion feels overly optimistic.
- Include a short section acknowledging known limitations or hurdles of implementing IDTEP.
- Briefly explain why tools like MATLAB, Unity3D, and Python were chosen — what unique features or benefits do they bring for students in this setting?
- Provide example results, such as student learning outcomes, project performance metrics, or qualitative feedback from pilot implementations, to strengthen claims about IDTEP’s effectiveness
- Ensure consistency when referring to the number of modules (six or seven?) and update the descriptions accordingly.
- While the section " Comparison of IDTEP with Other Educational Frameworks " mentions supporting studies (Monostori et al., Linn et al., Böhme et al.), it doesn’t provide concrete data, metrics, or experimental results showing how much IDTEP improves learning outcomes compared to other methods.
Round 2
Reviewer 2 Report
Comments and Suggestions for Authors
The authors have addressed all my comments in a convincing manner, and I recommend accepting this paper